# MiniGPT-Med: A Unified Vision-Language Model for Radiology Image Understanding

**Asma Alkhaldi**[1,2,3]*, **Raneem Alnajim**[2,3], **Layan Alabdullatef**[2,3], **Rawan Alyahya**[3], **Jun Chen**[2], **Deyao Zhu**[2], **Ahmed Alsinan**[3], **Mohamed Elhoseiny**[2]*

[1] *Johns Hopkins University (JHU)*

[2] *King Abdullah University of Science and Technology (KAUST)*

[3] *Saudi Data and Artificial Intelligence Authority (SDAIA)*

*aalkhal6@jhu.edu, mohamed.elhoseiny@kaust.edu.sa*

**Reviewed on OpenReview:** `https://openreview.net/forum?id=NenHFEg1Di`

## Abstract

Recent advances in artificial intelligence (AI) have precipitated significant breakthroughs in healthcare, particularly in the refinement of diagnostic procedures. However, existing studies have been limited in terms of functional coverage. This study introduces MiniGPT-Med, a vision-language model adapted from MiniGPT-v2 for medical applications through domain-specific fine-tuning on medical datasets. MiniGPT-Med demonstrates remarkable versatility across various imaging modalities, including X-rays, CT scans, and MRIs, enhancing its utility. The model is capable of performing tasks such as medical report generation, visual question answering (VQA), and disease identification within medical imagery. Its integrated processing of both image and textual clinical data markedly improves diagnostic accuracy. Our empirical assessments confirm the superior performance of MiniGPT-Med in disease detection, medical report generation, and VQA benchmarks, representing a significant step towards reducing the gap in assisting radiology practice. Furthermore, it achieves state-of-the-art performance in medical report generation, with substantial gains in BERT-Sim over both specialist and generalist baselines, improving by 17 and 12 points, respectively. MiniGPT-Med promises to become a unified Vision-Language model for radiology diagnoses, enhancing diagnostic efficiency across a wide range of medical imaging applications. Our model and code have been made publicly available on GitHub https://github.com/Vision-CAIR/MiniGPT-Med

## 1 Introduction

The unprecedented surge in both the quantity of image-text data across diverse fields and the strides made in vision-language modeling have paved the way for groundbreaking research in Generative Pretraining. This era of innovation is marked by the emergence of multimodal models such as GPT-4 Achiam et al. (2023) and Gemini Team et al. (2023). These advancements signify a leap forward in our ability to process and understand complex data. Despite this progress, the adoption of Multi-modal Large Language Models (LLMs) within the medical sector remains limited. The medical field's unique requirements for data complexity, sensitivity, and specificity highlight the need for tailored approaches to harness the potential of LLMs in transforming healthcare research and practice. Numerous models designed for medical applications have been introduced, yet they often exhibit a high degree of specialization for specific tasks. This specialization limits their versatility, particularly in performing diverse medical applications. For instance,

---

*Corresponding author.

models like Med-Flamingo Moor et al. (2023) and XrayGPT Thawkar et al. (2023b) are primarily tailored for tasks such as medical report generation and medical visual question answering, respectively. However, they lack capabilities in essential areas like disease detection, which requires visual grounding skills— a crucial component in the medical field. To address this deficiency, we introduce MiniGPT-Med, a unified model capable of adeptly handling both grounding and non-grounding tasks. We introduce MiniGPT-Med, a versatile model designed for various tasks in the medical domain, including but not limited to medical report generation, medical visual question answering, and disease identification. MiniGPT-Med builds upon the architecture of large language models (LLMs), which have demonstrated exceptional generative capabilities and extensive linguistics, including medical knowledge. Drawing on the successes of LLMs in a wide range of vision-language applications, as evidenced in recent studies Zhu et al. (2023); Chen et al. (2023); Li et al. (2024), our model adopts a design similar to MiniGPT-v2 Chen et al. (2023), utilizing the LLaMA-2 language model as a universal interface. Additionally, we incorporate distinct task identifiers to enhance the model's ability to accurately perform various medical vision-language skills. Through extensive experimentation, we have demonstrated that our model exhibits strong performance across a range of medical vision-language tasks, including medical report generation, medical visual question answering, and disease detection. We benchmarked our model against both specialized and generalized baseline models, revealing that our approach achieves strong results across all evaluated tasks. Notably, in the domain of medical report generation, our model attained state-of-the-art performance, surpassing the best baseline models by 19% in BERT-Sim and 5.2% in CheXbert-Sim. This indicates our model has strong generation capabilities on diverse medical vision-language tasks.

Our contributions are as follows:

1. We introduce MiniGPT-Med, a medical-domain adaptation of MiniGPT-v2 that retains the original architecture while extending it to heterogeneous radiological modalities, including X-rays, CT scans, and MRIs. MiniGPT-Med is fine-tuned on a diverse collection of medical datasets and supports multiple vision-language tasks—such as disease identification, medical visual question answering, and radiology report generation within a unified task identifier framework.

2. We conduct a comprehensive evaluation across grounding and non-grounding tasks, complemented by expert manual assessment. On medical report generation, MiniGPT-Med achieves substantial improvements in terms of BERT-Sim, outperforming the strongest specialist baseline by 17 points and exceeding the strongest generalist baseline (Gemini) by 12 points (72 vs. 60) under the same evaluation protocol.

## 2 Related Work

**Aligning Visual Data with Large Language Models:** Recent advancements in the domain of large language models, such as the release of GPT-4, have enhanced the interpretative and generative capabilities of LLMs. This progress is exemplified by models such as LLaVA Liu et al. (2024), Flamingo Alayrac et al. (2022), and MiniGPT-v2 Chen et al. (2023). LLaVA is designed to augment the understanding of visual content in large language models through diverse multimodal instructions. This enhancement in comprehension is critical for integrating different forms of data input. In contrast, Flamingo demonstrates remarkable proficiency in quick adaption to novel tasks with minimal data. This model effectively manages sequences that incorporate both visual and textual elements. MiniGPT-v2, on the other hand, displays enhanced multimodal capabilities within a singular model framework. This is achieved through task-specific training and a specialized architecture that combines visual tokens with a large language model, aligning well with the objectives of LLaVA and Flamingo.

While the aforementioned models primarily rely on supervised fine-tuning to align visual representations with LLMs, recent work has explored reinforcement learning (RL) as an alternative mechanism to encourage explicit reasoning. MedVLM-R1 Pan et al. (2025) introduces an RL-based medical vision–language model that generates natural language reasoning alongside final predictions for radiology visual question answering. Instead of relying on large-scale supervised datasets or chain-of-thought annotations,

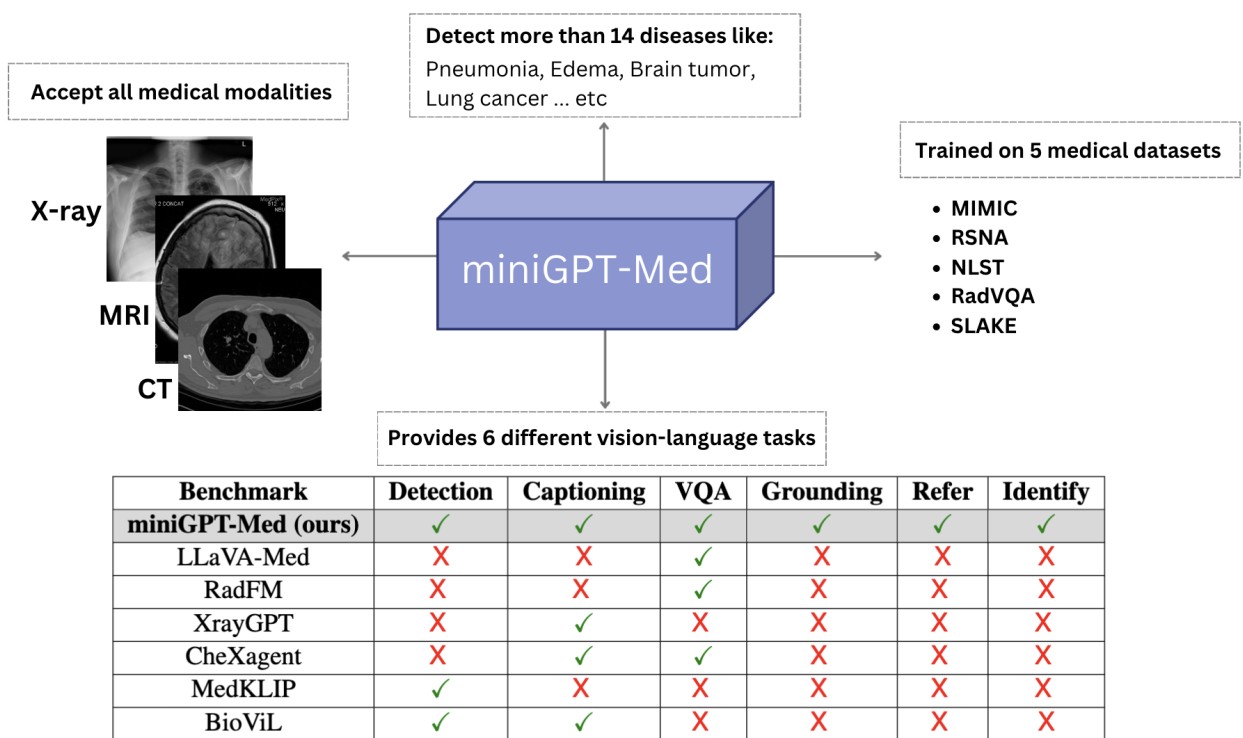

Figure 1: MiniGPT-Med demonstrates diverse capabilities across multiple clinical tasks, including disease detection, medical visual question answering, and medical report generation. It operates effectively on various radiological modalities such as X-rays, CT scans, and MRIs, and is capable of recognizing and diagnosing a broad range of conditions.

MedVLM-R1 employs Group Relative Policy Optimization (GRPO) to incentivize emergent reasoning using only final-answer supervision. Despite being trained on a limited number of samples and a compact backbone, the model demonstrates strong generalization across imaging modalities such as MRI, CT, and X-ray, outperforming larger supervised counterparts. This line of work highlights the potential of RL to enhance robustness in medical vision–language models beyond traditional supervised alignment strategies.

**Integration of Vision Language Models for Enhanced Medical Diagnostics:** Recent work in vision-language models has led to significant improvements in healthcare applications, especially in medical image analysis and diagnostic report generation. Utilizing VLMs in medical diagnostics marks a significant progression in the healthcare industry. Models combine computer vision and language processing to better analyze medical images like X-rays, computed tomography (CT), and MRIs. More specialized applications in the medical field such as LLaVA-Med Li et al. (2024) and Med-BERT Rasmy et al. (2020) have shown promise in incorporating structured electronic health records for improvements in disease prediction tasks. MedVQA Canepa et al. (2023) has demonstrated medical visual question-answering and image analysis capabilities. Furthermore, for classification and interpretation tasks, Med-Flamingo Moor et al. (2023), MedVis Shen et al. (2008), and MedMCQA Pal et al. (2022) have shown the importance of few-shot learning, visual interpretation, and domain-specific question-answering in medical AI. Both LLaVA-Med and Med-Flamingo focus on multimodal conversational AI and few-shot learning in medical contexts, utilizing large-scale datasets and showcasing proficiency in visual question answering. BioViL ban (2023), BioBERT Lee et al. (2019), and BioGPT Luo et al. (2022) all have tackled a more domain-specific language model pretraining. BioViL emphasizes text semantics for enhanced biomedical vision-language processing. Emphasis on specialized models for radiology applications has also been presented in MedKLIP Wu et al. (2023a), XrayGPT Thawkar et al. (2023a), and BERTHop Monajatipoor et al. (2021) all demonstrating the challenge of achieving high diagnostic accuracy. MedKLIP in particular innovates by integrating medical knowledge into vision-language

pre-training for improved disease classification. XrayGPT integrated a medical visual encoder with a large language model to combine visual and textual analysis to generate precise summaries from radiological data, while BERTHop showed diagnostic performance with smaller datasets on chest X-rays. Moreover, the contributions of CheXagent Chen et al. (2024), CheXNeXt Rajpurkar et al. (2018), and CheXpert Irvin et al. (2019) have set benchmarks in chest pathologies detection. While each work presents unique approaches, their common goal is to enhance radiological analysis through sophisticated AI models.

## 3 Method

### 3.1 Model architecture

Our model architecture, illustrated in Fig 2, is composed of three key components: a visual backbone, a linear projection layer, and an extensive language model. The details of each component are described as follows.

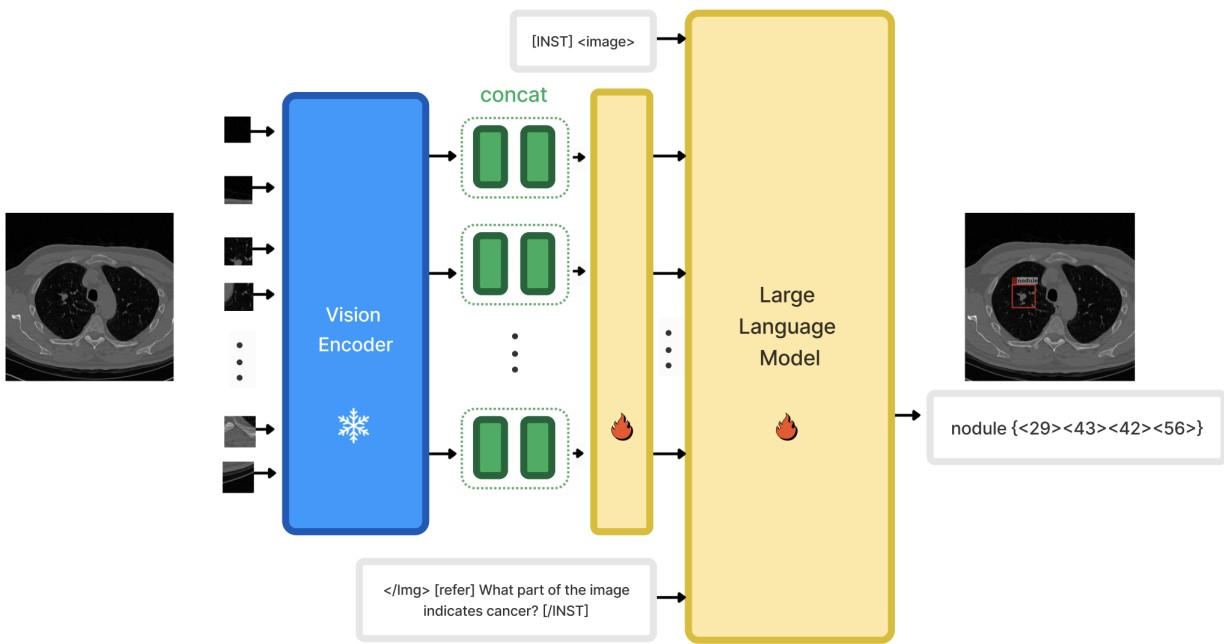

Figure 2: Overview of MiniGPT-Med Architecture: The model integrates a vision encoder, a linear projection layer, and a large language model (LLM). A single medical image is processed into visual semantic features via the pre-trained vision encoder, which is then transformed into a single visual token. This token is mapped to the LLM's space using the linear projection layer. During training, the vision encoder remains frozen, while the LLM and projection layer are fine-tuned for optimal performance.

**Vision Encoder.** We use EVA (Fang et al., 2022) as our vision backbone, which has been pre-trained on large-scale, publicly available datasets that capture essential visual representations such as shapes, patterns, and textures. To preserve the benefits of this pretraining and mitigate the risk of overfitting during fine-tuning, we initially freeze the vision encoder and fine-tune only the projection layer to ensure effective alignment with the textual modality. In cases where overfitting is observed, we explore unfreezing the encoder for further adaptation.

**Large Language Model (LLM).** Our model integrates LLaMA2-chat (7B) (Touvron et al., 2023), an opensource language model, as a unified interface for a broad range of medical vision–language tasks. Leveraging its flexible architecture, our system is capable of generating comprehensive medical reports and accurately localizing tumors in imaging data

Table 1: Task-specific instruction format. $<ImageFeature>$ denote the image features. During our model training, we used six different types of task identifiers for diverse grounding and non-grounding tasks.

| Task Types | [Identifier] Instruction |
|---|---|
| Caption | [caption] Could you describe the contents of this image for me? |
| VQA | [vqa] What plane is the image in? |
| Detection | [detection] pneumonia |
| Refer | [refer] the nodule in the left lung |
| Grounding | [grounding] describe this image in detail |
| Identify | [identify] what is this { <56><16><84><58>} |

**Vision-Language Alignment.** We enhance the model's efficiency by reducing the sequence length of visual tokens from the vision encoder, merging four adjacent tokens into a single embedding. This strategy helps maintain a manageable sequence length, enabling faster processing and more efficient computation. The resulting embeddings are then mapped into the language model's feature space through a linear projection layer. Importantly, this concatenation strategy does not sacrifice critical contextual information. Instead, it preserves spatial relationships and interactions among visual elements, which are essential for tasks like VQA.

## 3.2 Prompt Template

Our prompt template enables the model to handle diverse medical vision language tasks, including visual question answering, image captioning, referring expression comprehension (REC), referring expression generation (REG), disease detection, and grounded image captioning. To mitigate hallucinations and confusion common in handling various tasks, such as misidentifying lung tumors as calcifications in blood vessels or the heart, we incorporate task-specific tokens into the training framework. This ensures clarity and precision in multi-task environments. Our instruction design closely follows the template of MiniGPT-v2:

$$[INST]  < ImageFeature> </Img> [Task\ Identifier]\ Instruction\ [/INST]$$

We present diverse prompt templates in Table 1 to demonstrate how our model effectively deals with the different tasks through task identifiers.

## 3.3 Region grounding representation.

For tasks like disease detection and grounded image captioning, we represent bounding boxes as text, integrating spatial data into the language model. Coordinates are normalized to the [0,100] range and formatted as text. Each spatial location is expressed in the format:

$$\{< X_{left} >< Y_{top} >< X_{right} >< Y_{bottom} >\}$$

# 4 Experiments

## 4.1 Dataset Setup

The scarcity of quality medical datasets is a significant challenge in deep learning for medical imaging. To address this, we focused on curating a comprehensive radiology dataset, particularly for lung diseases and general medical information. Our collection includes diverse medical images (X-rays, CT scans, MRIs) and enriched datasets with bounding boxes, question-and-answer formats, and report generation to enhance model training. The datasets used include MIMIC Johnson et al. (2019), NLST The Cancer Imaging Archive (2023), SLAKE Medical Visual Question Answering (Med-VQA) (2023), RSNA Radiological Society of North America (2018), and RadVQA OSF (2023s), detailed as follows:

**MIMIC** MIMIC consists of 377,110 images and 227,835 medical reports. We used the XrayGPT Thawkar et al. (2023b) preprocessed MIMIC dataset, which includes 114,690 reports associated with 241,000 training images in JPG format. For our study, we split the data into 80% for training and 20% for testing, primarily for report generation.

**NLST** is used for lung cancer detection, containing 7,625 annotated low-dose CT scans with marked nodule locations. We extracted 2D CT slices from 3D volumes to highlight nodules. The annotations for training were sourced from Sybil Mikhael et al. (2023).

**SLAKE** used for grounding and VQA tasks, containing 579 radiology images of various organs and 3,543 question-answer pairs for training.

**RSNA.** We use the RSNA for evaluating pneumonia detection, consisting of 1,218 patients. We conduct a zero-shot evaluation on this dataset for disease detection.

**RadVQA** contains 315 radiology images across the head, chest, and abdomen, paired with 2,248 question-answer pairs spanning 11 categories. Responses are split between closed-ended (yes/no) and open-ended (short answers). We conduct zero-shot evaluation on this dataset.

### 4.2 Training Details

We initialized our model with pre-trained MiniGPT-v2 weights. During fine-tuning, we updated only the linear projection layer and applied LoRA Hu et al. (2021) to optimize the language model. The model was trained using cross-entropy loss and the AdamW optimizer. The dataset comprises 124,276 medical images with a resolution of 448×448 pixels, and no data augmentation was applied. Training was performed on a single NVIDIA A100 GPU for 100 epochs with a maximum learning rate of 1e-5. For hyperparameter tuning, we experimented with different epoch counts, learning rates, and optimization strategies. Evaluation was conducted every 20 epochs to select the best-performing checkpoint.

## 5 Evaluation

### 5.1 Baseline models

We evaluated MiniGPT-Med's performance on medical report generation, disease detection, and medical visual question answering (VQA), comparing it to specialist and generalist models. Specialist models handle either grounding or non-grounding tasks, while generalist models perform both. For the **medical report generation** task, we compared MiniGPT-Med with specialist models, including Med-Flamingo Moor et al. (2023) and LLaVA-Med Li et al. (2024), known for their prowess in vision language tasks and contextual learning abilities. Additionally, we compared MiniGPT-Med with RadFM Wu et al. (2023b), which is specifically tailored for radiology, and XrayGPT Thawkar et al. (2023b), a novel vision-language model designed for chest radiograph analysis. Furthermore, we evaluated MiniGPT-Med against CheXagent Chen et al. (2024), a foundation model focused on improving chest X-ray interpretation. Moreover, comparisons were made with generalist models like MiniGPT-v2 and Qwen-VL Bai et al. (2023), trained on the general vision-language data, showcasing exceptional performance across various vision-focused comprehension benchmarks. For the **disease detection** task, MiniGPT-Med was compared against specialist models including BioVil ban (2023), MedKLIP Wu et al. (2023a), and GLoRIA Huang et al. (2021), all pre-trained on vision-language medical datasets, as well as generalist models including MiniGPT-v2 and Qwen-VL. In the **medical VQA** task, we compared MiniGPT-Med with specialized models like MedVINT Zhang et al. (2023), OpenFlamingo Awadalla et al. (2023), and Med-Flamingo Moor et al. (2023) tailored to address the challenges of medical VQA, particularly in zero-shot scenarios, utilizing the RadVQA dataset. Additionally, our work was compared with generalist models such as MiniGPT-v2 and Qwen-VL to provide a comprehensive evaluation of MiniGPT-Med's performance.

## 5.2 Evaluation Metrics

In our study, we adapted our evaluation approach to align with the distinct skills required for interpreting radiology images using MiniGPT-Med. To assess the model's ability to generate radiological reports, we used two metrics: BERT Similarity (BERTsim) and CheXbert Similarity (CheXbert-Sim). BERTsim was utilized to evaluate the semantic similarity between the model-generated descriptions of radiological images and the expert-provided ground truth annotations. This involved using a BERT model to embed both the ground truth and generated sentences, followed by computing the cosine similarity between these embeddings. CheXbert-Sim, conversely, was selected for its relevance in assessing the model's accuracy in replicating professional medical report standards. It is a specialized version of the BERT model, fine-tuned on clinical texts, which computes the cosine similarity between embeddings for each corresponding sentence pair after encoding. For the Visual Question Answering (VQA) aspect, we exclusively used BERTsim to measure the semantic accuracy of the model responses. Additionally, we employed Intersection over Union (IoU) for grounding, a metric that quantitatively measures the model's precision in localizing and identifying specific features or abnormalities within the radiology images, such as pneumonia in the RSNA dataset.

## 5.3 Medical Report Generation

In our comprehensive study, we evaluate the efficacy of MiniGPT-Med for medical report generation on the MIMIC dataset Johnson et al. (2019). As summarized in Table 2, MiniGPT-Med consistently outperforms all specialist models, including the strongest domain-specific baseline, CheXagent, achieving improvements of 21.6 and 5.2 points on BERT-Sim and CheXbert-Sim, respectively. Among generalist models, MiniGPT-Med attains the highest BERT-Sim score, exceeding Gemini by 12 points, indicating stronger linguistic and semantic alignment with reference reports. In contrast, Gemini achieves the best performance on CheXbert-Sim, reflecting superior clinical concept consistency as measured by entity-based evaluation. These results highlight an inherent tradeoff between optimizing for linguistic fidelity (BERT-Sim) and clinical accuracy (CheXbert-Sim), suggesting that models excelling in natural language generation may not always achieve the highest scores on clinically grounded metrics, and vice versa.

Table 2: Evaluation of medical report generation on the MIMIC-CXR dataset. MiniGPT-Med is compared to generalist models capable of both grounding and non-grounding tasks, and specialist models limited to non-grounding tasks. The best scores for each model category are highlighted in **bold**.

| Method | Type | BERT–Sim | CheXbert–Sim |
|---|---|---|---|
| MedFlamingo | | 10.4 | 3.2 |
| LLaVA-Med | | 6.2 | 17.5 |
| RadFM | Specialist | 45.7 | 17.5 |
| XrayGPT | | 44.0 | 24.2 |
| CheXagent | | 50.4 | 24.9 |
| MedVLM-R1 | | **55.0** | **27.0** |
| Gemini | | 60.0 | **57.4** |
| GPT-4o | | 58.0 | 55.0 |
| MiniGPT-v2 | Generalist | 53.0 | 21.1 |
| Qwen-VL | | 51.9 | 20.3 |
| **Ours** | | **72.0** | 30.1 |

## 5.4 Disease Detection

The data showcased in Table 3 reveal that MiniGPT-Med stands out for its competitive performance when compared against a comprehensive range of baseline models. With an Intersection over Union (IoU) score of 0.26, MiniGPT-Med not only exceeds the capabilities of generalist models by a margin of 16% but also attains performance metrics on par with specialist models. The peak IoU score among these specialist models is noted to be 0.31. Our MiniGPT-Med achieves competitive results and it demonstrates good disease detection

performance among all the baseline models, highlighting its potential as a versatile and effective tool in the medical domain.

Table 3: Evaluation of disease detection on the RSNA benchmark in a zero-shot setting: comparison of our model with generalist and specialist models. The best performance is highlighted in **bold**.

| Method | Model Type | RSNA IoU |
| --- | --- | --- |
| BioViL | | 0.30 |
| MedKLIP | Specialist | **0.31** |
| GLoRIA | | 0.21 |
| Qwen-VL | | 0.10 |
| MiniGPT-v2 | Generalist | 0.13 |
| **Ours** | | **0.26** |

## 5.5  Medical Visual Question Answering

This study evaluates our model, MiniGPT-Med, against various baseline models using the RadVQA OSF (2023s) benchmark, as presented in Table 4. MiniGPT-Med achieves a notable performance metric of 0.58, surpassing both generalist models such as MiniGPT-v2 Chen et al. (2023) and specialist models like OpenFlamingo Awadalla et al. (2023) and Med-Flamingo Moor et al. (2023). This performance not only demonstrates MiniGPT-Med's superiority over a broad range of models but also shows it can achieve results comparable to those of the leading specialist model, MedVIN Zhang et al. (2023), which has an accuracy of 0.62. The ability of MiniGPT-Med to outperform or match the performance of several specialized and generalist models underscores its significant potential as a foundation for the development of advanced medical visual question-answering models.

Table 4: Evaluation of visual question answering on the RadVQA benchmark in a zero-shot setting, using BERT-Sim as the evaluation metric. Comparison includes generalist and specialist models, with top performance in each category highlighted in **bold**.

| Method | Model Type | RadVQA BERT-Sim |
| --- | --- | --- |
| MedVIN | | **0.62** |
| OpenFlamingo | Specialist | 0.49 |
| Med-Flamingo | | 0.48 |
| Qwen-VL | | 0.13 |
| MiniGPT-v2 | Generalist | 0.55 |
| **Ours** | | **0.58** |

## 5.6  Assessing Hallucinations

Hallucination is a common issue for large language models such as ChatGPT and Gemini, which sometimes generate inaccurate content. Our model, equipped with the LLaMA language model, may also occasionally produce hallucinated medical reports. As there is no established automatic method for detecting hallucinations in the biomedical domain, we rely on expert radiologists to review the generated reports and quantify the level of hallucination in the model outputs. We asked three senior radiologists to assess 50 randomly sampled cases from the MIMIC dataset, focusing on the robustness, granularity, and accuracy of the generated reports. The evaluation was conducted using three questions: Q1) how closely the generated report aligns with the radiologist's expert judgment, Q2) the level of clinical detail in the report, and Q3) the accuracy of the report for diagnosing pathologies. Each response was labeled as Good, Medium, or Poor. For each model, we aggregate all radiologist judgments across the three questions and normalize the counts of each label by the total number of evaluations, yielding the average radiologist evaluation percentages reported in

Table 5. We apply the same evaluation protocol to competing methods, including XrayGPT and CheXagent, to ensure a fair comparison.

Table 5: Expert evaluation of medical report generation. The table shows the percentage of expert votes for each quality category (Good, Medium, Poor) across three models.

| Quality | MiniGPT-Med | CheXagent | XrayGPT |
|---------|-------------|-----------|---------|
| Good | 65.48% | 61.78% | 49.00% |
| Medium | 29.33% | 23.33% | 21.79% |
| Poor | 5.19% | 16.43% | 27.67% |

The analysis shows that 65.48% of the medical reports generated by our model are of high quality and better than CheXagent and XrayGPT. Furthermore, 29.33% of the reports are classified as medium quality, and only 5.19% are poor, which is the lowest rate among the compared models. This distribution underscores the model's ability to produce reports that meet professional standards by offering detailed and accurate diagnostic information. These findings highlight MiniGPT-Med's considerable potential as a reliable and effective tool for advancing medical reporting processes.

# 6 Qualitative Examples

To illustrate the versatility of our proposed model, we present qualitative examples in multiple clinical vision-language tasks. These include detection, anatomical grounding, region identification, referring expressions, VQA, and medical report generation. As shown in Figure 3, the model accurately interprets diverse input modalities and responds to task specific prompts. These examples highlight the model's strong generalization capabilities and its potential to support various real world diagnostic.

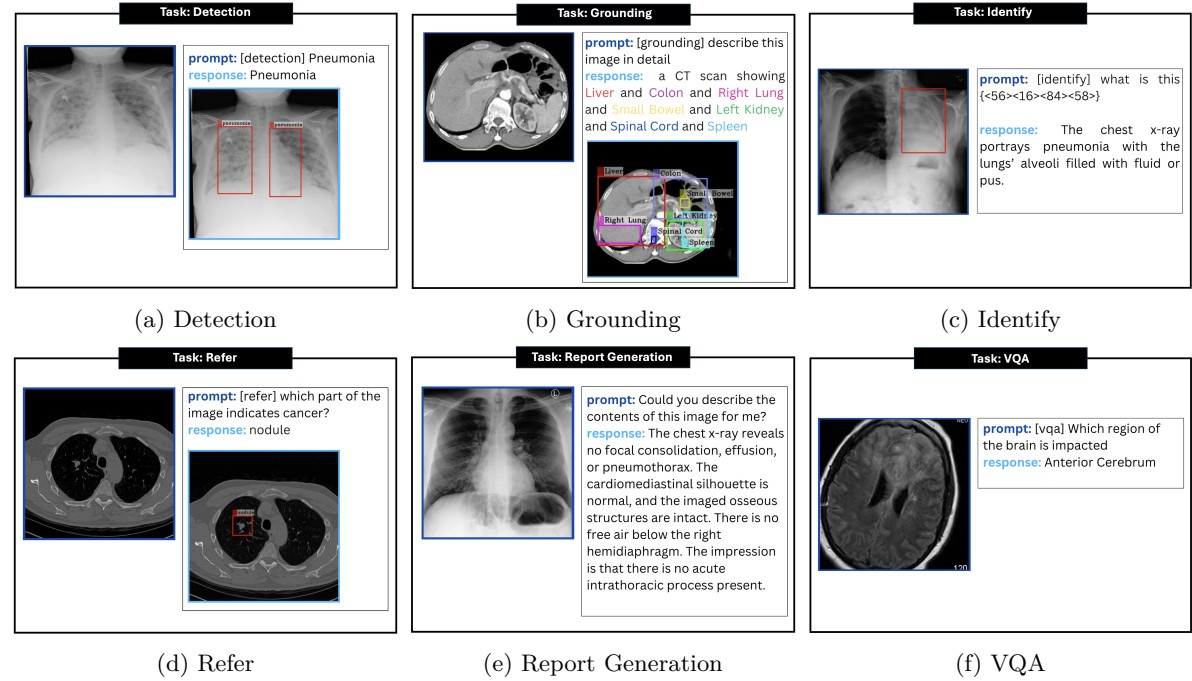

Figure 3: Examples of MiniGPT-Med multi-task abilities, including: (a) disease detection, (b) anatomical grounding, (c) region identification, (d) referring expressions, (e) medical report generation, and (f) visual question answering.

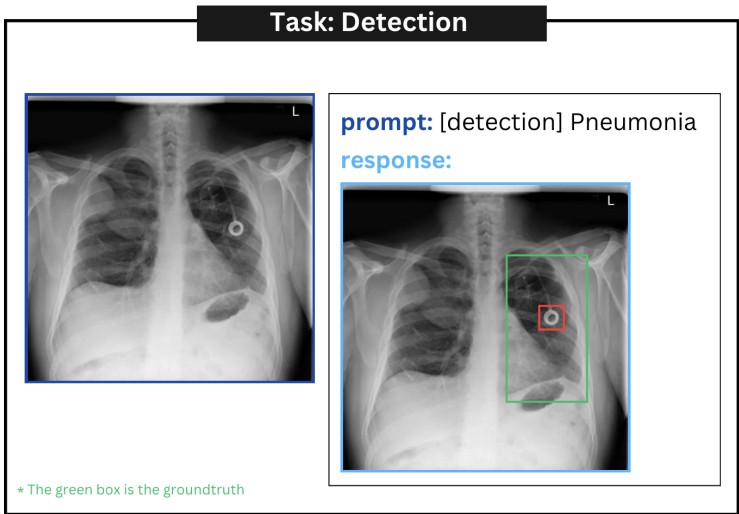

Figure 4: False positive example. The red bounding box indicates a falsely detected disease, while the green bounding box shows the ground truth.

## 7 Limitation

Despite its notable capabilities, MiniGPT-Med faces challenges stemming from a lack of diverse and high-quality training datasets. This limitation restricts the model's coverage to a narrow range of diseases. To overcome this, the expansion and diversification of training datasets are crucial, enabling the integration of extensive medical knowledge into the model. Additionally, MiniGPT-Med occasionally experiences issues with generating inaccurate medical reports and improperly connecting symptoms to diseases, a phenomenon known as hallucination. These problems are partly due to the inadequate quality of annotated data available for training. Addressing these issues requires not only richer datasets but also the incorporation of advanced vision backbones and enhancements in the underlying large language model to ensure more accurate and reliable outputs. Additionally, the deployment of deep learning models in the health sector has encountered several notable failure cases. One such example for MiniGPT-Med is their difficulty distinguishing between the abnormality and the medical images that include device implants in the human body. Fig. 4 demonstrates a sample of the data that MiniGPT-Med failed in correctly identifying the pneumonia location. The object under the green bounding box is the ground truth and the object under the red bounding box is the false detection. The model easily confuses the device implants as an abnormality. This shortcoming often results in misdiagnosed conditions. Specifically, when AI encounters X-rays or MRIs featuring implants, it may incorrectly identify these as abnormalities.

## 8 Conclusions

In this study, we introduce MiniGPT-Med, a specialized multi-modal designed for radiology diagnosis applications. It handles various medical vision-language tasks such as generating medical reports, detecting diseases, and answering visually-based medical questions, using distinct task identifiers to navigate these tasks efficiently. MiniGPT-Med outperforms baseline models in both grounding and non-grounding tasks, achieving state-of-the-art performance in the MIMIC-CXR medical report generation task. Radiologist evaluations show that approximately 65% of the generated reports are of preferred quality, highlighting the model's superiority. Future plans include incorporating more diverse medical datasets, improving the understanding of complex medical terminology, enhancing interpretability and dependability, and conducting extensive clinical validation studies to ensure effectiveness and safety in real healthcare environments.

## Appendix

## A  Patient-Level Data Splits

All datasets used in our experiments were partitioned at the patient ID level to prevent any overlap of subjects between the training, validation, and test sets. This ensures that no patient appears in more than one split, thereby eliminating the risk of information leakage across splits.

For each dataset, patient IDs were randomly assigned to the training, validation, and test subsets according to predefined split ratios. Once assigned, all samples associated with a given patient (e.g., images, volumes, or reports) were included exclusively in the corresponding split. This protocol was applied consistently across all datasets and tasks considered in this work.

## B  NLST Slice Selection Protocol

The NLST dataset consists of chest CT scans provided as 3D volumetric images. To enable 2D-based model training and evaluation, a consistent slice selection protocol was applied. Each NLST scan includes the annotated nodule location specified in 3D coordinates (x,y,z). For each scan, we extracted the axial slice corresponding to the z-coordinate of the annotated nodule center. This slice typically provides the highest visual prominence of the target nodule and aligns with standard clinical practice for nodule assessment. The selected slice was used as the representative 2D input for both training and evaluation.

## C  Preprocessing Pipeline Specifications

To ensure fair and controlled comparisons, we standardized the preprocessing pipeline across all models, tasks, and datasets, including all baselines.

**Image Preprocessing**

- All images were converted to grayscale.

- Images were resized to $448 \times 448$ using bilinear interpolation with antialiasing.

- Pixel intensities were normalized using a shared mean and standard deviation.

**Data Augmentation**

- During training, RandAugment was applied.

- During evaluation, no data augmentation was used.

## D  Ablation Studies

We conduct ablation experiments to analyze the impact of key architectural and training design choices on performance and efficiency in the medical report generation, disease detection, and VQA tasks.

### D.1  Effect of Model Components.

Table 6 reports the effect of removing or modifying individual components of our model. We evaluate performance on MIMIC (report generation), RSNA (disease detection), and RadVQA (VQA).

| Experiment | MIMIC | RSNA | RadVQA |
|---|---|---|---|
| | (BERT-Sim) | (IoU) | (BERT-Sim) |
| **Our Model** | **0.72** | **0.26** | **0.58** |
| Removing instruction token | 0.72 | 0.02 | 0.58 |
| With Q-Former | 0.72 | 0.22 | 0.58 |
| Token-merging (2→1) | 0.66 | 0.18 | 0.54 |
| Token-merging (None) | 0.66 | 0.18 | 0.54 |
| Unfreezing vision encoder | 0.57 | 0.15 | 0.47 |

Table 6: Ablation study evaluating the impact of architectural and training design choices on medical report generation (MIMIC), disease detection (RSNA), and visual question answering (RadVQA).

Removing the instruction token leads to a severe degradation in detection performance on RSNA, indicating that explicit task identifiers are critical for structured prediction tasks. Token merging and unfreezing the vision encoder consistently degrade performance across all tasks, suggesting that aggressive compression or full visual fine-tuning may harm multimodal alignment under limited data. Incorporating a Q-Former does not yield consistent gains, highlighting that additional architectural complexity does not necessarily translate to improved performance in this setting.

## D.2 Efficiency and Resource Trade-offs.

Table 7 analyzes the computational cost of different design choices in terms of training time, memory usage, and inference latency.

| Experiment | GPUs | Train Time | FLOPs | VRAM | Latency | Speedup |
|---|---|---|---|---|---|---|
| | | (hrs) | (rel.) | (GB) | (ms) | (vs. Base) |
| **Baseline** | **1×A100** | **20** | **1.00×** | **22** | **85** | **1.00×** |
| Remove instruction token | 1×A100 | 20 | 1.00× | 22 | 85 | 1.00× |
| With Q-Former | 2×A100 | 26 | 1.15× | 25 | 100 | 0.77× |
| Token-merging 4→2 | 1×A100 | 20 | 1.00× | 22 | 85 | 1.00× |
| Token-merging 2→1 | 2×A100 | 40 | 1.60× | 28 | 125 | 0.50× |
| Token-merging (None) | 4×A100 | 60 | 2.20× | 36 | 170 | 0.33× |
| Unfreeze vision encoder | 2×A100 | 27 | 1.00× | 22 | 85 | 0.74× |

Table 7: Computational cost and efficiency trade-offs of different model variants in terms of training time, memory usage, and inference latency.

Some variants increase computational cost, requiring more GPUs, longer training time, and higher inference latency. In contrast, the baseline provides a good balance between performance and efficiency.

## E Prompt Templates

We report below the full set of prompt templates used for each task. We use the same prompts across all datasets and evaluation splits, and show them exactly as used. Curly braces `{}` denote variable placeholders (e.g., spatial coordinates or regions).

### E.1 Report Generation

- `Describe this image in detail.`

- `Take a look at this image and describe what you notice.`

- `Please provide a detailed description of the picture.`

- `Could you describe the contents of this image for me?`

### E.2 Grounding

- `[grounding] please describe this image in detail`

- `[grounding] describe this image as detailed as possible`

- `[grounding] summarize this image in detail`

- `[grounding] give a thorough description of what you see in this image`

### E.3 Detection

- `[detection] pneumonia`

- `[detection] nodules`

- `[detection] abnormality`

- `[detection] tumor`

### E.4 Referring Expression (Refer)

- `[refer] tumor`

- `[refer] abnormality`

- `[refer] where is the tumor?`

- `[refer] where is pneumonia?`

- `[refer] where is the nodule?`

- `[refer] give me the location of the tumor`

- `[refer] give me the location of the abnormality`

- `[refer] could you tell me the location for pneumonia?`

- `[refer] from this image, tell me the location of the tumor`

- `[refer] from this image, tell me the location of the abnormality`

- `[refer] from this image, tell me the location of the nodules`

### E.5 Object Identification

- `[identify] {}`

- `[identify] what object is in this location {}`

- `[identify] identify the object present at this location {}`

- `[identify] what is it in {}`

- `[identify] describe this object in {}`

- `[identify] this {} is`

- `[identify] the object in {} is`

### E.6 Visual Question Answering (VQA)

For the VQA task, the model accepts both closed-form and open-ended natural language questions. No fixed prompt template is enforced; questions are provided directly in free-form text during both training and evaluation.

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
