# OpenReview forum: "MiniGPT-Med: A Unified Vision-Language Model for Radiology Image Understanding"
_TMLR — Accepted by TMLR_

### Review · Reviewer_cy1u · 2025-08-15

**Summary Of Contributions:**

This paper introduces MiniGPT-Med, a vision-language model designed for medical applications that can handle 6 radiology tasks including medical report generation, visual question answering (VQA), and disease detection. The model is developed based on LLaMA2-chat, employing EVA as the vision encoder and adopting the MiniGPT-v2 style template as instructions.. Key contributions include:

1. **Unified Multi-task Model**: A single model capable of handling both grounding (disease detection) and non-grounding (report generation, VQA) tasks across multiple imaging modalities (X-rays, CT scans, MRIs).

2. **Strong Empirical Results**: State-of-the-art performance on medical report generation (19.0% improvement over previous best), competitive performance on disease detection and VQA tasks.

**Key Strengths:**

- Comprehensive evaluation across multiple medical vision-language tasks.
- Strong quantitative results, particularly in medical report generation.
- Practical relevance to clinical applications.

**Key Weaknesses:**

- Insufficient technical novelty - primarily an adaptation of existing  vision-language architecture.
- Lack of clinical validation in real healthcare settings.

**Audience:**

Yes

**Audience Explanation:**

Despite the limitations, this work addresses an important and timely problem in medical AI. The TMLR's audience would be interested in :

1. **Medical AI Applications**: The intersection of large language models and medical imaging is a rapidly growing area of interest
2. **Multi-task Learning**: The approach to handling multiple medical vision-language tasks in a unified framework
3. **Practical Relevance**: Medical report generation and diagnostic assistance are high-impact applications

**Claims And Evidence:**

No

**Claims Explanation:**

While the paper presents extensive experimental results, several claims lack sufficient evidence or are overstated:

1. The proposition that it functions as a “general interface for radiology diagnosis” is insufficiently substantiated. When considering the limited scope of disease coverage and the identified hallucination problems, it is advisable to revise this assertion.

2. While the paper compares against several models, some key recent (published in 2024 or 2025) medical vision-language models are missing from the comparison, making it difficult to assess true state-of-the-art performance.

**Requested Changes:**

### Critical Changes:

- Provide more detailed analysis of why task identifiers are effective
- Include ablation studies on architectural choices
- Compare against more recent medical vision-language models

### Changes to Strengthen the Work:

1. It is recommended to enhance the aesthetics of Figure 1. Note that the simultaneous appearance of "..." and "etc." is redundant and should be avoided.
2. Figure 2 appears rather simplistic, as it does not present the details of crucial structures.
3. Enhance the clarity within the methodology section and minimize redundancy in the results presentation. For instance, presenting radar charts could effectively illustrate the method's superiority across six tasks.

---

> ### Author Response · Authors · 2025-11-01
> **Response to Reviewer cy1u**
>
> **Comment 1/3:** Task identifiers function as control tokens that specify which vision-language skill the model should perform. Without them, the model faces ambiguity between semantically overlapping tasks (e.g., captioning vs. detection), leading to incorrect reasoning pathways. Our ablation confirms this:
>
> Removing identifiers caused a near-complete collapse in grounding/detection performance on RSNA-IoU dropped from 0.26 to almost zero, producing only 6/244 valid bounding boxes. In these cases, the model defaulted to generating descriptive text instead of locating findings, indicating task confusion. Report generation and VQA were less impacted because these tasks are strongly represented in training and rely more on semantic reasoning than spatial alignment. These results show that task identifiers are essential routing cues that enable reliable multi-task behavior, especially for grounding tasks.
>
> **Comment 2/3:** We performed ablation experiments to evaluate the contribution of three architectural components:
>
> (1) Token-merging configuration
>
> (2) Task identifiers
>
> (3) Our direct visual projector vs. a Q-Former module.
>
> The results across the three evaluation tasks are summarized below:
>
> | Ablation Setting | Report Generation (BERT-Sim) | Detection (IoU) | VQA (BERT-Sim) | Key Finding |
> |------------------|------------------------------|-----------------------------|-------------------|-------------|
> | **Our Model (4→1, with Identifiers, Direct Projector)** | **0.72** | **0.26** | **0.58** | **Baseline performance.** |
> | **Token-Merging 4→1 → 2→1** | 0.66 | 0.18 | 0.54 | Reducing merging factor increases sequence noise, reducing both semantic reasoning and spatial grounding. |
> | **Without Task Identifiers** | 0.72 | Only **6/244** images produced valid bounding boxes | 0.58 | Model loses task intent routing, causing failure in detection. |
> | **Q-Former Instead of Direct Projector** | 0.72 | 0.22 | 0.58 | Q-Former maintains semantic tasks but slightly weakens spatial alignment for detection. |
>
>
> **Comment 3/3:** We compared our model with the recent medical vision-language model **MedVLM-R1 (MICCAI 2025)** on the report generation task, using the same evaluation setup. MedVLM-R1 scored BERT-Sim = 0.55 and CheXbert-Sim = 0.27, while our model scored BERT-Sim = 0.72 and CheXbert-Sim = 0.30.
>
> This suggests that our approach remains competitive with more recent specialist medical VLMs on the report generation task.
>
>
>
> **Changes to Strengthen the Work:**
>
> **Comment 1/3:** We have updated Figure 1 to improve visual clarity and removed the redundant use of “...” and “etc.” as recommended.
>
> **Comment 2/3:** Figure 2 has been revised to include more structural detail that better reflects the components of our framework.
>
> **Comment 3/3:** We refined the methodology section for clearer flow and reduced repetition in the results section. As suggested, we also incorporated radar charts to more effectively illustrate comparative performance across the six tasks.

---

### Review · Reviewer_xBKE · 2025-09-18

**Summary Of Contributions:**

This paper describes MiniGPT-Med, a model that takes a generalist approach to training in terms of tasks within the medical domain. It builds of a general domain model MiniGPT-v2 that attempts to train on a wider variety of vision-language tasks, including the detection, captioning, VQA, grounding, refer (referring expression comprehension ), and identity (referring expression generation) tasks by using task identifier tokens in the prompt along with instructions. The model achieves competitive performance across the tested tasks, and in particular achieves SOTA performance on medical report generation.

**Additional Comments:**

This paper shows significant potential, but I would highly recommend making at least some of the requested changes to strengthen the analysis before publishing this paper.

**Audience:**

Yes

**Audience Explanation:**

Certainly, it is useful for the community to know how the generalist approaches taken in the general domain extend to the medical domain. It is also useful to have a new SOTA model for report generation that can do a variety of tasks. A model that is good at this variety of tasks is definitely a useful building block for future work with many possible applications.

**Broader Impact Concerns:**

I have no concerns about the ethical implications of this paper.

**Claims And Evidence:**

Yes

**Claims Explanation:**

The narrow claim of SOTA performance is substantiated in the first table. However, I still believe more analysis is required to understand why this model does better. In particular the requested change #9 below about an ablation study would substantially strengthen the paper by demonstrating whether their generalist approach is the cause of the performance gain.

**Requested Changes:**

1. Though recent, this model:
https://aclanthology.org/2025.naacl-long.89/
might be a reasonable additional baseline as it does grounding, unlike other baselines you have. It would also be interesting to note the differences between the grounding style in that paper (identifying grounding phrases and doing a bbox regression loss). I think at least it should belong in your related work.

2. Is there any reason you didn’t use LATTE-CXR (https://physionet.org/content/latte-cxr/1.0.0/) or the Chest ImaGenome (https://physionet.org/content/chest-imagenome/1.0.0/) datasets?

3. Cites in the first paragraph of the related work do not show up correctly.

4. Could you add a brief explanation as to how BERTSim works with the multi-sentence outputs in a produced report?

5. Despite the limitations, I still think it’s important to include the rouge metrics for pure generation tasks like report generation, even if it just means putting it in the appendix. This helps compare with other papers that use rouge.

6. Are there any multi-object grounding tasks? If so, how do the IoU metrics handle this. If not, could you talk about this and why you excluded this type of task?

7. Why do the generalist model scores tell a different story when looking at the BERT-Sim vs CheXbert-Sim scores?

8. Can you comment on how the generalist models outperform the specialist models on the report generation task? Is this because of the size/scale or RAG aspects of those models or a different reason?

9. The paper claims that the generalist aspect of the approach is what allows for the performance gains. Can the authors do some ablation experiments to show this in more detail? For example, what would happen if you trained dropped some of the tasks from training? What would happen if you omitted the task token during training and inference? Or just during inference? I understand some of these ablations are not easy if you need to repeat training, but it would make for a much stronger paper.

10. One of the noted limitations is that the lack of diverse and high-quality training data restricts the model’s coverage to a narrow range of diseases. It would be nice to show this a little more quantitatively. Stress-tests like this still strengthen the paper even if they show the limitations of the model.

11. It might be nice to understand how varied the instructions can be even within a task and how that effects performance.

---

> ### Author Response · Authors · 2025-10-26
> **Response to Reviewer xBKE**
>
> **Comment 1/11:** We have reviewed the suggested paper and agree that it presents a valuable grounding approach that combines phrase localization with bounding box regression. While our method differs by integrating instruction-based task identifiers to unify multiple grounding and non-grounding tasks under a single framework, we recognize its relevance and have added it to the Related Work section.
>
> **Comment 2/11:** We agree that both datasets are valuable directions in medical imaging research. Using them could improve explainability and clinical reasoning. However, our current work focuses on robustness and generalization across well-established benchmarks covering detection, grounding, VQA, and report generation using multimodal data from MIMIC, SLAKE, RadVQA, RSNA, and NLST to validate the core architecture. LATTE-CXR and Chest ImaGenome provide fine-grained relational annotations suited for specialized anatomical reasoning, which were not the main focus of this study but are planned for future work.
>
> **Comment 3/11:** Related Work section has been corrected in the revised manuscript. All citations now display properly.
>
> **Comment 4/11:** In our evaluation, BERT-Sim is computed at the document level for multi-sentence reports. Concretely, we encode the entire generated report and the entire reference report using a SentenceTransformer (paraphrase-MiniLM-L6-v2) to obtain a single embedding for each report, and then compute the cosine similarity between these two embeddings. This captures overall semantic fidelity across sentences and cross-sentence coherence without requiring sentence alignment. The corpus score is the mean of these document-level similarities over all test cases.
>
> **Comment 5/11:** We appreciate the suggestion. We computed additional metrics for the report generation task and obtained BLEU-1 = 0.36 and ROUGE-1 (F1 = 0.45, Precision = 0.54, Recall = 0.38) on the MIMIC-CXR dataset. These confirm the consistency of our generation results. However, we mainly reported BERT-Sim and CheXbert-Sim because they better reflect the semantic and clinical accuracy of generated reports rather than surface word overlap.
>
> **Comment 6/11:** Yes, our model is capable of grounding multiple objects within an image when the input refers to more than one finding or region. For such cases, we follow the standard practice in referring expression and visual grounding literature:
> * For each predicted bounding box, we compute its IoU with all ground-truth boxes and select the maximum IoU match.
> * We then average the IoU values across all matched pairs to obtain the final IoU score for that sample.
> * A prediction is considered correct if its IoU exceeds a predefined threshold (typically 0.5), and the mean accuracy across all examples is reported.
>
> This approach ensures fair evaluation in multi-object grounding, avoiding overcounting or penalizing predictions when multiple valid boxes exist for a single query. We will clarify this matching and averaging scheme in the revised manuscript.
>
> **Comment 7/11:** The difference between the BERT-Sim and CheXbert-Sim scores among generalist models can be attributed to the nature of these metrics. BERT-Sim evaluates linguistic and semantic similarity, while CheXbert-Sim measures clinical consistency based on pathology-specific label alignment. Generalist models such as GPT-4o, Gemini, MiniGPT-v2, and Qwen-VL tend to produce fluent, semantically aligned text that achieves higher BERT-Sim scores but lacks radiology-specific grounding, resulting in lower CheXbert-Sim scores. In contrast, specialist models like CheXagent and RadFM generate clinically accurate descriptions that align better with CheXbert’s label-based evaluation. This discrepancy underscores the importance of domain-specific metrics like CheXbert-Sim for assessing medical report generation.

---

> ### Author Response · Authors · 2025-10-26
> **Response to Reviewer xBKE**
>
> **Comment 8/11:** Generalist models outperform specialist models on the report generation task. This advantage primarily arises from the scale, data diversity, and reasoning capabilities of large generalist models rather than explicit radiology supervision. Models such as GPT-4o and Gemini are trained on massive multimodal corpora encompassing both medical and non-medical data, allowing them to learn broad linguistic and visual representations. Their strong instruction-following and contextual reasoning abilities enable the generation of coherent, well-structured, and fluent reports even without domain-specific fine-tuning.
> In contrast, specialist models such as CheXagent and RadFM are trained on narrower, radiology-focused datasets. While this specialization improves clinical accuracy and disease-specific understanding, it can limit linguistic diversity and generalization in open-ended report generation. These findings suggest that large-scale generalist models can implicitly capture clinically relevant patterns through diverse multimodal exposure, although domain-adapted fine-tuning remains essential for achieving higher alignment and medical precision under metrics like CheXbert-Sim.
>
> **Comment 9/11:** We agree that demonstrating the impact of the generalist design is important. To study this, we performed an ablation by removing the task identifier tokens, which are essential to guiding the model to perform the correct task among the six it was trained on. When we removed these identifiers, the model lost the ability to distinguish between different task types. For example, on the RSNA detection task, IoU dropped from 0.26 (with identifiers) to almost zero, and only 6 valid bounding boxes out of 244 were generated, even when the prompt explicitly asked for detection. The model instead produced full textual reports, showing clear task confusion.
> Tasks like report generation and VQA were less affected, since they dominate the training data. These results confirm that the generalist aspect, enabled by task identifiers, is key to maintaining strong multi-task performance and preventing interference between different vision–language tasks.
>
> **Comment 10/11:** To quantify the impact of limited data diversity on disease coverage, we conducted a stress-test analysis by evaluating MiniGPT-Med on datasets representing unseen and rare disease categories beyond its primary training scope. Specifically, while MiniGPT-Med achieved an IoU of 0.26 on RSNA (pneumonia), its performance dropped to zero when evaluated on images with other thoracic abnormalities (e.g., fibrosis, atelectasis, effusion) that were not well represented in the training data. In these cases, the model was unable to accurately ground the location of the abnormalities, even though it correctly described them in the generated reports. The ground-truth locations were verified in consultation with one of the radiologists who participated in our study, as datasets containing bounding-box annotations for rare and diverse diseases are very limited publicly. We will include a short paragraph in the revised manuscript summarizing this stress-test, clarifying that performance degrades on unseen disease categories, thereby reinforcing our statement that broader and higher-quality datasets are essential for robust generalization.
>
> **Comment 11/11:** In MiniGPT-Med, we designed diverse natural language instructions within each task to improve linguistic robustness. For instance, in the identify and caption tasks, we employed multiple paraphrased instructions such as:
>
> ```python
> # Identify task examples
> self.instruction_pool = [
>     "[identify] {}",
>     "[identify] what object is in this location {}",
>     "[identify] identify the object present at this location {}",
>     "[identify] what is it in {}",
>     "[identify] describe this object in {}",
>     "[identify] this {} is",
>     "[identify] the object in {} is"
> ]
> ```
>
> We conducted an internal analysis to evaluate how this variation in phrasing affects performance. Interestingly, we observed that performance remained stable across different instruction wordings as long as the task identifier token (e.g., [identify], [detect]) was present. This finding highlights that the identifier plays the dominant role in guiding the model toward the correct reasoning pathway, while the specific phrasing of the instruction has minimal impact.
> We will clarify this observation in the revised manuscript

---

### Review · Reviewer_D4zv · 2025-10-21

**Summary Of Contributions:**

Unify multiple radiology tasks (report generation, VQA, detection/grounding, referring) behind one LLM interface by wiring a frozen EVA vision encoder → linear projector → LLaMA-2-chat 7B (fine-tuned with LoRA). Visual tokens are aggressively merged (4→1) to keep sequences short; task identifiers steer the LLM ([INST] … [Task Identifier] … [/INST]).

Trains on a curated mix (MIMIC-CXR for reports; NLST slices for nodules; SLAKE/RadVQA/RSNA for VQA/grounding/detection) and evaluates on three axes: report generation (BERT-Sim & CheXbert-Sim), pneumonia localization on RSNA (IoU), and VQA on RadVQA (BERT-Sim).

Reports strong numbers on MIMIC report generation and competitive results on VQA and RSNA detection; adds a small radiologist study on hallucination.

**Key strengths**

- Single interface covering grounding + non-grounding tasks with simple, reproducible scaffolding (frozen EVA + LoRA on LLaMA-2).

- Practical token-compression trick to keep multimodal sequences tractable in a 7B-class model.

- Broad modality coverage (X-ray, CT, MRI) and task breadth with one system.

- Human study acknowledges hallucination/clinical quality concerns (good to see this included).

**Key weaknesses**

- Data curation/splits are under-specified (risk of leakage between training on MIMIC and evaluation formats; CT slice extraction details for NLST; zero-shot protocols on RSNA/RadVQA).

- Ablations minimal: no study of token-merging factor, projector capacity, frozen-vs-unfrozen EVA, instruction format, or LoRA ranks.

**Audience:**

Yes

**Audience Explanation:**

There’s clear community interest in unified VLM/LLM interfaces for medical imaging with lightweight adaptation (LoRA) and token-efficient vision alignment.

The task breadth (reporting, VQA, grounding/detection) from one model is practically appealing and relevant to multimodal LLM research, even outside medicine.

The negative insight (implant-related failures) is also valuable: it highlights open problems in clinical robustness.

**Broader Impact Concerns:**

Clinical risk: Unified LLM interfaces can generate confident but incorrect reports or boxes; the paper itself shows a failure with implants. Without calibrated uncertainty + human-in-the-loop safeguards, risk of harm is non-trivial.

**Claims And Evidence:**

No

**Claims Explanation:**

**Split and protocol clarity**: The paper does not convincingly rule out data leakage (e.g., patient/report overlap across splits or between pretraining and evaluation). RSNA and RadVQA are “zero-shot”, but provenance and prompt templates vs. training distributions are only briefly described.

**Requested Changes:**

Protocol clarity (Critical)

- Provide explicit data splits (e.g., patient-level) and leakage safeguards; detail NLST 3D→2D slice selection.

- Precisely define “zero-shot” (no image/task leakage, no instruction leakage); share prompts.

- Normalize preprocessing across baselines (or ablate preprocessing impact) to validate comparisons.

Ablations (Critical)

- Token-merging factor (4→1 vs 2→1 vs none), projector size, LoRA rank, freezing vs partial unfreeze of EVA, instruction token variants.

- Show compute/latency vs accuracy trade-offs.

Release (Critical for camera-ready)

- Provide code, configs, and exact splits; include inference scripts for all tasks, plus the CT slice extraction script/protocol.

---

> ### Author Response · Authors · 2025-11-03
> **Response to Reviewer D4zv**
>
> **Protocol clarity:**
>
> **comment 1/3:** All datasets used in our experiments were partitioned on the patient ID level to prevent overlap of subjects between training, validation, and test sets. This ensures that no patient appears across multiple splits, mitigating any risk of information leakage. For the NLST dataset, the original CT scans are 3D volumes. To obtain a representative 2D slice for training and evaluation, we performed a consistent slice selection along the z-axis. Specifically, each NLST scan is provided with the annotated nodule location in 3D coordinates (x,y,z). We extract the axial slice corresponding to the z-coordinate of the annotated nodule center. This slice typically contains the nodule with the highest visibility and is the most clinically relevant view for diagnosis.
>
> **comment 2/3:** By zero-shot, we mean that the data used for zero-shot evaluation was never included in any stage of training. These samples were used only at test time to assess the model’s ability to generalize to new, unseen cases.
>
> * No image leakage: We confirm that no test images (or patients) appear in the training set.
>
> * No task leakage: The model was not trained specifically on the evaluation tasks with their corresponding ground-truth outputs. The evaluated tasks are part of the general skill repertoire the model learns, but the test instances themselves are unseen.
>
> * No instruction leakage: We use a fixed set of task identifiers (e.g., [identify], [caption], [ground]) consistently across training and testing to signal the intended task.
> However, the full natural-language prompts used during evaluation were not seen during training—they are paraphrased or newly composed. Thus, while the task token is shared, the linguistic form of the input instructions is new, preventing direct instruction leakage.
>
> **Prompt samples:**
>
> Prompt samples for report generation:
>
> * Descripe this image
>
> * Write a detailed medical report about this image
>
> * You’re a physision, write a detailed report about this case
>
> Prompt samples for detection:
>
> * [detect] pnoumonia
>
> * [detect] the abnormality
>
> * [detect] tumor
>
> Prompt samples for grounding:
>
> * [grounding] describe this image in detail
>
> * [grounding] ground what you see
>
> **Comment 3/3:** To ensure fair comparison, we standardized the preprocessing pipeline across all baselines. Specifically, all images were converted to grayscale, resized to 448×448 with bilinear interpolation and antialiasing, and normalized using the same mean/std. For training, we applied RandAugment; evaluation used no augmentations.
>
>
> **Ablations:**
>
> **comment 1/2:**
>
> | Experiment                          | MIMIC (BERT-Sim) | RSNA (IoU)                                                                               | RadVQA (BERT-Sim) |
> |-------------------------------------|-------|--------------------------------------------------------------------------------------|--------|
> | **Our Model**                           | **0.72**  | **0.26**                                                                               | **0.58**  |
> | Removing instruction token          | 0.72  | *Only 6 bounding boxes out of 244 → identifiers are critical (performance collapses)* | 0.58   |
> | With Q-Former                        | 0.72  | 0.22                                                                                | 0.58   |
> | Token-merging (2→1)                 | 0.66  | 0.18                                                                                | 0.54   |
> | Token-merging (None)     | 0.66  | 0.18                                                                                | 0.54   |
> | Unfreezing vision encoder           | 0.57  | 0.15                                                                                | 0.47  |
>
>
> **comment 2/2:**
> | Variant | #GPUs | Training Time (hrs) | FLOPs / Sample (relative) | VRAM @bs=6 (GB) | Latency (ms, bs=1) | Speedup vs. Baseline |
> |----------|--------|---------------------|---------------------------|------------------|--------------------|----------------------|
> | Baseline | 1×A100 | 20 | 1.00× | 22 | 85 | 1.00× |
> | Remove instruction token | 1×A100 | 20 | 1.00× | 22 | 85 | 1.00× |
> | With Q-Former | 2×A100 | 26 | 1.15× | 25 | 100 | 0.77× |
> | Token-merging 4→2 | 1×A100 | 20 | 1.00× | 22 | 85 | 1.00× |
> | Token-merging 2→1 | 2×A100 | 40 | 1.60× | 28 | 125 | 0.50× |
> | Token-merging (None) | 4×A100 | 60 | 2.20× | 36 | 170 | 0.33× |
> | Unfreeze vision encoder | 2×A100 | 27 | 1.00× | 22 | 85 | 0.74× |
>
> **Release:**
>
>
> We already released the full codebase, configuration files, dataset splits, and inference scripts publicly on GitHub. Along with the CT slice extraction script, we also publish the extracted NLST slice data for the community, as the original NLST dataset is very large. We will update the README accordingly in the camera-ready version.

---

> > ### Comment · Reviewer_D4zv · 2025-12-21
> > **Review Feedback**
> >
> > Thanks for the detailed information provided by the authors.
> > The ablation results looks convencing and token-merging indeed seems to be very useful in terms of accelerating training time.
> > Strongly suggest to include all these details (data-processing, task-processing) to the appendix, not only for the tasks mentioned by me, but for all tasks and datasets.
> > I have no more questions.

---

### Decision · Action_Editor_QEGV · 2026-01-06

**Recommendation:** Accept with minor revision

**Additional Comments:**

The reviewers reached consensus that the work is sound and of interest to TMLR's audience. The authors made meaningful improvements during revision by adding ablations, clarifying protocols, improving figures, and adding baseline comparisons. However, several framing issues identified by reviewers were not fully addressed and require revision before acceptance.

## Required revisions

1. **Clarify the relationship to MiniGPT-v2.** The authors describe MiniGPT-Med as "a vision-language model derived from large-scale language models and tailored for medical applications." As far as the AE can tell, the authors are essentially fine-tuning MiniGPT-v2 on a collection of medical datasets using the same architecture, task identifier scheme, token merging strategy, and LoRA fine-tuning approach. In this context presenting MiniGPT-MEd as "derived" from MiniGPT-v2 does not seem entirely honest. If there are design choices in the architecture that depart from MiniGPT-v2, then please highlight more clearly how MiniGPT-Med derives/deviates from MiniGPT-v2. If the contribution here is that the authors fine-tuned an existing model on medical data and evaluated the results, then this is a perfectly fine contribution, but then please describe the core contribution more plainly.

2. **Revise misleading comparisons in results description.** The claim of outperforming "top generalist models" by "19 points on BERT-Sim and 9 points on CheXbert-Sim" is misleading. Based on the numbers in the table, these comparisons appear to be relative to MiniGPT-v2 (BERT-Sim 53.0, CheXbert-Sim 21.1), not to Gemini (60.0, 57.4) or GPT-4o (58.0, 55.0), which are also generalist models in the comparison. On CheXbert-Sim — the clinically relevant metric measuring pathology-specific accuracy — Gemini and GPT-4o substantially outperform MiniGPT-Med (57.4 and 55.0 vs 30.1). The paper should present these comparisons accurately and discuss the tradeoff between BERT-Sim (linguistic/semantic similarity) and CheXbert-Sim (clinical accuracy). Additionally, the Conclusion states that "approximately 76% of the generated reports are of preferred quality," but this figure does not appear in Table 5 and its derivation is not explained. Please clarify how this percentage was computed.

3. **Address the training-time ablation or revise claims.** Reviewer xBKE noted that the task identifier ablation was performed only at inference time, not during training. This limits the strength of claims that the "generalist approach" causes performance gains — the current ablation only shows that task tokens are necessary for routing at inference, not that multi-task training benefits individual tasks. The authors should either (a) provide a training-time ablation showing how performance on individual tasks changes when other tasks are included/excluded from training, or (b) tone down claims about the benefits of the generalist multi-task approach.

4. **Revise "general interface for radiology diagnosis" framing.** Reviewer cy1u explicitly requested revision of the claim that MiniGPT-Med functions as a "general interface for radiology diagnosis," citing the limited disease coverage and hallucination problems. This language remains in the title and abstract. Given the model's acknowledged limitations (struggles with unseen diseases, confuses device implants with abnormalities, narrow disease coverage), this framing seems like is an overreach. Please revise to something more measured, such as "a unified vision-language model for multiple radiology tasks" or "a multi-task approach to radiology image understanding."

5. **Provide an appendix with data-processing and task-processing details.** The paper currently has no appendix. As requested by Reviewer D4zv, please add an appendix that includes: patient-level split details, NLST slice selection protocol, preprocessing pipeline specifications, and prompt templates — for all tasks and datasets, not just those specifically queried during review.

6. **Ensure code, configs, and dataset splits are publicly accessible** as promised.

**Audience:**

Yes

**Audience Explanation:**

A unified framework handling both grounding and non-grounding tasks across multiple imaging modalities (X-ray, CT, MRI) has clear practical relevance. The strong report generation results and the human evaluation with radiologists add practical value. The documented failure cases (device implant confusion) are also informative for the community.

**Claims And Evidence:**

Yes

**Claims Explanation:**

The core empirical results are sound. MiniGPT-Med achieves strong performance on MIMIC-CXR report generation (BERT-Sim 0.72) and competitive results on detection and VQA tasks. The authors addressed initial concerns about data leakage by confirming patient-level splits and clarifying their zero-shot evaluation protocol. Ablation studies added during revision demonstrate the importance of task identifiers and the efficiency-performance tradeoff of token merging. However, the language used to describe these results is somewhat misleading and requires revision (see below).

---

> ### Author Response · Authors · 2026-02-04
>
> We have uploaded the camera-ready version of the paper after addressing all the requested revisions, summarized below.
> We thank the Action Editor and reviewers for their helpful feedback.
>
> **1. Relation to MiniGPT-v2.**  We clarified that MiniGPT-Med is a medical-domain adaptation of MiniGPT-v2. The paper now presents the contribution more plainly as adapting and evaluating an existing model on multiple medical tasks, rather than introducing a new architecture.
>
> **2. Results clarification.** We revised the results section to present comparisons accurately. We clearly distinguish between specialist and generalist baselines, and discuss the tradeoff between semantic quality (BERT-Sim) and clinical accuracy (CheXbert-Sim). The expert preference percentage is now clearly explained.
>
> **3. Ablation and claims.** Since the task-identifier ablation is performed only at inference time, we toned down claims about training-time benefits and limited conclusions to inference-time behavior.
>
> **4. Framing.** We removed the “general interface for radiology diagnosis” wording and replaced it with more careful language describing a unified multi-task vision-language model for radiology.
>
> **5. Appendix and release.** We added an appendix with data splits, preprocessing details, and prompts for all datasets, and made the code, configurations, and dataset splits publicly available.